# Deceptive orbital confinement at edges and pores of carbon-based 1D and 2D nanoarchitectures

Ignacio Piquero-Zulaica [1,6] ✉, Eduardo Corral-Rascón[1,6], Xabier Diaz de Cerio [2,6], Alexander Riss [1] ✉, Biao Yang [1], Aran Garcia-Lekue [2,3] ✉, Mohammad A. Kher-Elden[4], Zakaria M. Abd El-Fattah [4], Shunpei Nobusue [5], Takahiro Kojima[5], Knud Seufert [1], Hiroshi Sakaguchi [5] ✉, Willi Auwärter [1] & Johannes V. Barth[1]

The electronic structure defines the properties of graphene-based nanomaterials. Scanning tunneling microscopy/spectroscopy (STM/STS) experiments on graphene nanoribbons (GNRs), nanographenes, and nanoporous graphene (NPG) often determine an apparent electronic orbital confinement into the edges and nanopores, leading to dubious interpretations such as image potential states or super-atom molecular orbitals. We show that these measurements are subject to a wave function decay into the vacuum that masks the undisturbed electronic orbital shape. We use Au(111)-supported semiconducting gulf-type GNRs and NPGs as model systems fostering frontier orbitals that appear confined along the edges and nanopores in STS measurements. DFT calculations confirm that these states originate from valence and conduction bands. The deceptive electronic orbital confinement observed is caused by a loss of Fourier components, corresponding to states of high momentum. This effect can be generalized to other 1D and 2D carbon-based nanoarchitectures and is important for their use in catalysis and sensing applications.

The bottom-up on-surface chemistry route facilitates the production of graphene nanostructures such as covalent organic frameworks[1,2], nanographenes[3–5], and graphene nanoribbons[6] (GNRs) that are difficult to achieve via traditional solution chemistry. These nanostructures are promising for next generation nanoelectronic, spintronic and optoelectronic devices[7], as well as, photodetectors[8], membranes as smart filters and sensors for biological and chemical sensing applications[9,10]. Their versatility is based on the fact that the electronic properties are largely determined by their structural characteristics including size and edge structure, which can be exclusively accessed on the atomic level with scanning tunneling microscopy/spectroscopy (STM/STS).

Conductance features observed in the d$I$/d$V$ (or STS) maps of graphene-based nanostructures such as GNRs can be correlated with the local density of states (LDOS) of their corresponding valence bands (VBs) and conduction bands (CBs) wave functions. However, Söde et al. found that the LDOS associated with the VBs and CBs of the seven-carbon-atom-wide armchair-GNR (7-AGNR) was most intensely observed along the nanoribbon edge in STS[11]. Depending on the wave function oscillations parallel and perpendicular to the growth

[1]Physics Department E20, TUM School of Natural Sciences, Technical University of Munich, James-Franck-Straße 1, D-85748 Garching, Germany. [2]Donostia International Physics Center (DIPC), Paseo Manuel de Lardizabal 4, E-20018 Donostia-San Sebastian, Spain. [3]Ikerbasque, Basque Foundation for Science, 48013 Bilbao, Spain. [4]Physics Department, Faculty of Science, Al-Azhar University, Nasr City E-11884 Cairo, Egypt. [5]Institute of Advanced Energy, Kyoto University, Uji 611-0011 Kyoto, Japan. [6]These authors contributed equally: Ignacio Piquero-Zulaica, Eduardo Corral-Rascón, Xabier Diaz de Cerio. ✉e-mail: ipiquerozulaica@gmail.com; a.riss@tum.de; wmbgalea@ehu.eus; sakaguchi@iae.kyoto-u.ac.jp

direction of the 7-AGNR, some orbitals would decay faster into the vacuum. It was elucidated that certain orbitals would concentrate more strongly at the edges of the 7-AGNR due to a lack of cancellation of positive and negative regions of the wave function along the edges[11]. Therefore, d$I$/d$V$ maps at constant-height mode, which correlate with the squared modulus of the wave functions at a fixed $z$ that corresponds to the tip-sample distance, ask for a detailed analysis of the wave function decay from the carbon lattice. Since then, the LDOS intensity confinement into the GNR edges has been widely observed in chevron GNRs[12], topological GNRs[13,14], zigzag GNRs[15], notched ("Bite" defect staggered) 8-AGNRs[16], and boron-doped GNRs[17] to name a few. It should be noted that for instance the edge states of topological chiral-GNRs[18] and the zigzag end states of substrate-decoupled finite 7-AGNRs[19] are inherently localized and therefore experimentally observed at the edges as well.

Theoretical studies predicted that the electronic properties of GNRs could be strongly modulated by the insertion of nanopores[20,21]. Recently some GNRs with holes have been synthesized on surfaces[22–26] and only for a few of them the nature of the electronic properties was explored[26–28]. For the latter cases, the experimental d$I$/d$V$s and the theoretical LDOS maps show that the spectroscopic features corresponding to the VBs and CBs appear most prominent at the GNR outer edges and in its nanopores[26–28].

The electronic properties of nanographenes are also strongly dependent upon the insertion of nanopores, enabling the engineering of novel functionalities. Nanoporous nanographenes in the form of coronoids[29–32], open-shell triangulenes[3], triangulenes with open nanopores[5], and nanographenes with azulene moieties[33] have been synthesized. In all these cases, intense conductance features located at the nanographene edges and nanopores have been observed in d$I$/d$V$ maps, but many different interpretations have been given. While some attribute these conductance features to the CO-functionalization of the tip[31], in other cases they are also present for metal tips[3]. Some

studies claim that LDOS simulations cannot predict the edge and nanopore confined features[31], however, other studies correctly capture them[3,32]. In several cases these edge and nanopore confined electronic orbitals have been correlated with electronic wave functions whose weight is located at the carbon backbone (i.e., corresponding to HOMO, LUMO, negative or positive ion resonances)[3,5,31,32], while, in other cases, they have been assigned to super-atom molecular orbitals (SAMO) whose wave functions are intrinsically localized at the nanopores but are lying at very high energies[33].

A similar scenario should also be expected for nanoporous graphene (NPG) structures. Atomically precise NPGs can be fabricated via lateral fusion of bottom-up on-surface synthesized GNRs[27,34–39]. In particular, Moreno et al. synthesized an NPG with 1 nm size pores by laterally fusing notched 7-13-AGNRs[35]. Indeed, in d$I$/d$V$ maps performed at moderate voltages (≈2 V), conductance features were observed localized at the nanopores and NPG edges. The origin of these confined states was however attributed to free electron-like image potential states (IPS) lying high in energy, constricted at the vacuum side along the GNR edge and localized most prominently at the nanopores.

From the above literature, it is evident that a clear and general understanding of such edge and nanopore localized conductance features in different types of GNRs, nanographenes and NPGs is still pending (see schematic Fig. 1a). For this purpose, it is desirable to study this effect both on individual notched GNRs and on NPG built by their lateral coupling. In this work, we use the 4',5"-dibromo-1,1':2',1":2",1"'-quaterphenyl (DBQP) as the building block to form, by subsequent annealing steps, well-defined gulf-type GNRs (g-GNRs) and NPG on Au(111) in ultra-high vacuum (UHV) conditions (see Fig. 1b). The electronic properties of the structures obtained in each step are characterized using STM/STS, non-contact atomic force microscopy (nc-AFM) with a CO-functionalized tip and complemented with density functional theory (DFT) and electron plane-wave expansion (EPWE)

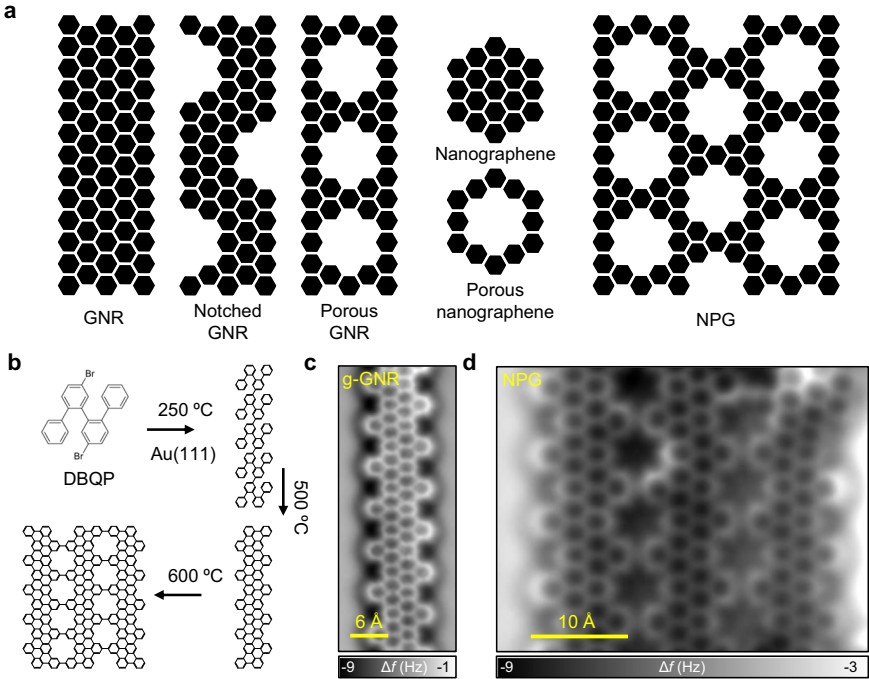

**Fig. 1 | Examples of non-porous and porous GNRs, nanographene, graphene-like nanostructures, and the g-GNR and NPG used in this study. a** Schematic illustration of different types of GNRs, nanographenes and NPG where the electronic orbital confinement at the edges and pores should be apparent in STS. **b** Synthetic path for the generation of g-GNRs and NPG structures. The DBQP precursor is deposited on Au(111) held at 250 °C leading to debromination and coupling to form oligomer chains. After annealing at 500 °C, an intramolecular cyclodehydrogenation leads to the formation of planar g-GNRs. Finally, annealing at 600 °C leads to the lateral fusion of a few g-GNRs and atomically precise NPG patches are formed. **c, d** Nc-AFM images of the respective products, i.e., g-GNR and NPG on Au(111).

simulations[40]. This study provides a generalized understanding of the electronic orbital decay into the vacuum and its confinement into the edges and nanopores of carbon-based 1D and 2D nanoarchitectures, which is a common effect present in STS measurements on such systems. We identify the effect, which is enhanced with increasing the probe-sample separation, to originate from a loss of Fourier components corresponding to states of high momentum.

## Results

### On-surface synthesis of g-GNRs and NPG

The choice of Au(111) as a catalyst surface provides an on-surface reaction path with well-separated thermal windows that leads to a hierarchical control of the DBQP molecule transformation into oligomers, g-GNRs and NPG[27,35,36] (see Fig. 1b). The DBQP precursor monomers deposited on Au(111) held at 250 °C, are thermally driven to first form polyphenylene oligomers via dehalogenation and subsequent radical combination[41]. The estimated length of the oligomers is up to 20 nm, similar to other cove-type GNRs[42] (see Supplementary Fig. 1).

The oligomer chains consist of C-C coupled monomers with a periodicity of ≈0.74 nm. Since the reaction temperature is above the threshold of C-C coupling (190 °C on Au(111))[43–45], no organometallic phase is expected[46]. Similarly to chevron and chiral-GNRs, oligomers tend to show tilted phenyls that appear as bright spots when imaged in STM[43,47]. Those tilted phenyl groups most probably engage in π-π interactions, resulting in the alignment of adjacent polymer chains and their bunching into stable close-packed islands[6,34,47] (see Supplementary Fig. 1).

At 500 °C the oligomers undergo intramolecular cyclodehydrogenation and form g-GNRs (Fig. 1c). Unlike the oligomers, that exhibit tilted phenyl rings in STM, nc-AFM imaging of the g-GNR using a CO-terminated tip shows a flat and uniform structure, indicating its complete planarization. A unit cell periodicity of ≈0.74 nm along the g-GNR axis is obtained. Due to the high formation temperatures required, branching of g-GNRs and the emergence of defects along the nanoribbon backbone are quite common. Nanoribbon cross-coupling, segment flipping, and rearrangements of carbon atoms are most likely responsible for the entangled structures observed (see Supplementary Figs. 2 and 3 for more details on the defects created). Despite this scenario, it is still possible to find regular g-GNR segments ranging from 4 to 12 unit cells in length, bearing similar electronic characteristics to infinite ribbons[48] (Supplementary Figs. 4–8).

Employing a large packing density of g-GNRs (i.e., close to full monolayer) leads to the formation of cross-linked g-GNRs into NPGs at 600 °C (Fig. 1d). Such structures arise from the cyclodehydrogenative coupling of two or more adjacent g-GNRs[47] whereby NPGs ranging from two to five fused g-GNRs can be achieved in a very low yield (see Supplementary Fig. 11). Nc-AFM images show the atomic structure of the g-GNR fusing sites and the resulting pores of the NPG (Fig. 1d). In some fused g-GNR regions, the graphene lattice structure remains unperturbed and atomic precision is conserved[47] (see Supplementary Figs. 13 and 14). The g-GNRs laterally fuse with the phenyl rings bonded through the *meta* positions and the rings lay flat on the surface[34]. The formation of a single-phase NPG is achieved, for which a g-GNR zipping mechanism is suggested (see Supplementary Fig. 12). The resulting H-passivated nanopores are distributed in periodic out-of-phase arrays with a high areal density of ≈$1.3 \times 10^{14}$ pores per cm[2] and represent one of the smallest planar pores that can be fabricated in graphene (≈5.7 Å in diameter). In summary, the notched shape of g-GNRs and the small nanopores of NPG achieved serve as interesting and adequate templates to induce edge and nanopore confined conductance features in STS, thereby allowing us to study the origin of these peculiar electronic effects in detail.

### Electronic properties of g-GNRs

Figure 2 analyzes the electronic properties of a g-GNR segment of five unit cells in length. A series of d$I$/d$V$ point spectra are shown in Fig. 2a,

which have been taken at specific g-GNR positions (e.g., edge gulfs, edge phenyls, and ribbon center)[12,34,49,50] using a CO-functionalized tip. We find that the STS line shapes significantly differ from that of the Au(111) substrate (gray spectrum), which is practically featureless except for the surface state onset at ≈−460 mV[51]. We can identify a very strong increase of intensity at 1.9 V that is most intense at the gulf positions (red spectra), attenuates at the edge phenyls (blue spectra) and appears very dim at the ribbon center (orange spectrum). The LDOS is strongly asymmetric for unoccupied and occupied states whereby the latter states appear much less intense. However, some weak peaks can still be identified between −1.0 V and −1.5 V, which are located at the lateral phenyl rings and gulfs (blue and red spectra). Such significant intensity differences between occupied and unoccupied states may stem from the strongly varying probabilities for tunneling into or out of these states[31].

To understand the origin of the STS peaks, we capture the spatial distribution of the LDOS by acquiring a series of constant-height d$I$/d$V$ maps at selected energies (Fig. 2b). For the occupied region, the intensity is mainly localized at the lateral phenyl rings, appearing as two bright dots at −1.1 V and winding along the edges at −1.3 V. In the unoccupied region (at 1.9 V and 2.1 V), a highly localized conductance feature is present at the gulfs, notably protruding laterally outside the ribbon. Recently, similar conductance features have been observed in chevron GNRs[52], holey nanographenes[3,32,33], molecular nanoporous networks[53], molecular arrays[54], and NPGs[35]. However, as mentioned above, the origin attributed for such unoccupied states is very diverse, ranging from surface state confinement[52], IPS[35], LUMO states[3,5,32], NIR[31], SAMO states[33] or a superposition of non-interacting states[54].

To clarify the origin of these states, extensive DFT calculations were performed. The band structure of g-GNRs displayed in Fig. 2d shows a wide (1.84 eV) bandgap, with dispersive and braiding bands for VBs and CBs along the longitudinal g-GNR direction (i.e., ΓY). In addition, at higher energies, an IPS band can also be identified (see Supplementary Fig. 9). Such IPS originates from the free-electron-like states that are confined at the vacuum side along the g-GNR edge[35]. Figure 2c shows wave functions at the Γ point extracted for VB-1, VB, CB, and CB + 1. The morphology of the experimental VBs shows some resemblance with the calculations, but the experimentally observed features appear to be markedly at the edges (compare for instance the VB wave function with d$I$/d$V$ map at −1.1 V). However, the DFT-derived CBs wave functions, which do not show any contribution in the gulf regions, are drastically different from the experimental observations for unoccupied d$I$/d$V$ maps.

To shed light on this matter, we perform DFT-based LDOS map simulations in the Tersoff-Hamann approximation (considering a s-wave tip) for VBs and CBs. As shown in Fig. 2e, a good match with the d$I$/d$V$ maps is found for a tip height of $z = 5$ Å. To account for experimental broadening, an energy integration of 100 meV and 250 meV is required for occupied and unoccupied states, respectively (see light pink integration regions in the band structure of Fig. 2d). The higher integration region required for the CBs might be attributed to a possible hybridization of CBs with Au(111)[55]. Note that even though a CO-tip is used for the d$I$/d$V$ maps, the set-point current is rather low and therefore the predominant orbital of the tip is s-wave[31]. This is confirmed by the resemblance of the CO-tip d$I$/d$V$ maps with those acquired with a metal tip as shown in Supplementary Fig. 8. We can now confidently assign the frontier peaks at −1.2 V and 1.9 V to the onsets of delocalized VBs and CBs, and exclude their relation to the IPS and the SAMO. According to the calculations, it is unlikely for IPS or SAMO states to be detected at such low bias voltages (≈2 V), and in fact they were observed beyond 3.7 V on graphene nanoflakes and $C_{60}$ molecules on Au(111)[56,57]. The bandgap size of the g-GNR is thus ≈2.8 V which is further corroborated by d$I$/d$V$ spectra taken on lines across edges and the ribbon center (see Supplementary Fig. 4). Since the bandgap is expected to depend on the GNR length[58], we have repeated

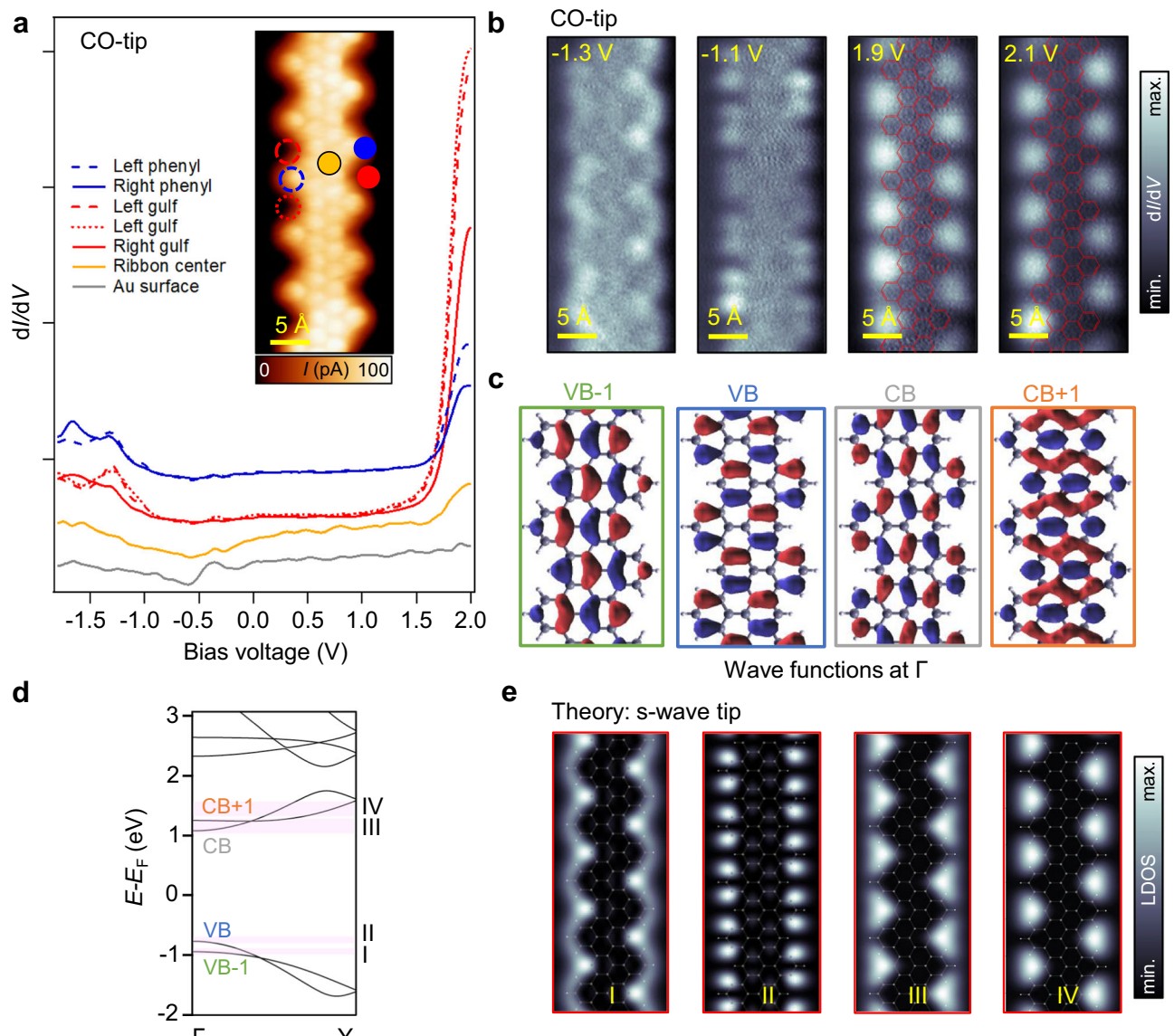

**Fig. 2 | Electronic properties of g-GNRs. a** d$I$/d$V$ point spectra ($I$ = 100 pA; $V$ = 2.0 V, $f$ = 289 Hz; $V_{mod\ (0\text{-to-peak})}$ = 30 mV) acquired at right (solid) and left (dashed) gulf positions (red), phenyls (blue) and at the ribbon center (orange), as well as the pristine Au(111) substrate (gray). A CO-functionalized tip is used. A wide bandgap of ≈2.8 V is observed between the onsets of the VB and CB. The inset shows a bond-resolved STM image ($V$ = −8.3 mV) obtained by using a CO-functionalized tip in constant-height mode. **b** Constant-height d$I$/d$V$ maps acquired near the VB [tip stabilization parameters: ($V$ = −1.3 V, $I$ = 40 pA) and ($V$ = −1.1 V,

$I$ = 40 pA)] and CB [tip stabilization parameters: ($V$ = 1.9 V, $I$ = 50 pA) and ($V$ = 2.1 V, $I$ = 60 pA)] frontier orbitals reveal a very rich substructure most prominently localized at the ribbon edges ($f$ = 289 Hz; $V_{mod\ (0\text{-to-peak})}$ = 40 mV). **c** Wave functions at the Γ point for VB, VB-1, CB, and CB + 1. **d** DFT electronic band structure for the g-GNR showing dispersive and braiding VBs and CBs and a semiconducting gap of 1.84 eV. **e** DFT-based LDOS map simulations at different energies close to the onsets of the VBs and CBs obtained at a height of $z$ = 5 Å away from the g-GNR plane.

these measurements on longer g-GNRs, yielding a similar value for the bandgap (see Supplementary Fig. 7), thus excluding significant finite-size effects. For completeness, and to rule out any possible influence of the CO-tip, we also measure the g-GNR bandgap with a metal tip and obtain the same result (see Supplementary Fig. 8). The bandgap size mismatch between theory and experiment (1.84 eV $vs$ 2.8 eV) can be attributed to the absence of a substrate in the calculations as well as to the well-known limitation of DFT that tends to underestimate VB-CB gaps by several hundreds of meV[59]. Nevertheless, the convincing experiment-theory agreement allows us to unambiguously determine a wide bandgap for the g-GNR/Au(111), in contrast to the previous experimental assignment of a narrow bandgap of 1.1 eV[37].

Interestingly, d$I$/d$V$ maps show similar gulf confined conductance features for all positive voltages explored, i.e., up to 3 V (further increase of the samples bias compromises CO-tip and g-GNR integrity,

see Supplementary Figs. 7 and 8). DFT-based LDOS map simulations show that indeed all CBs show this pattern, which is coincidentally very similar to the IPS that appears at even higher energies (see Supplementary Fig. 10). Since the simulations are performed without the substrate, the Au Shockley surface state confinement[52] can also be ruled out as a possible origin of the experimentally observed localization in the gulf region, which can thus be unambiguously attributed to the CBs (Supplementary Fig. 9). However, the details of the physical mechanism behind the emergence of such deceptive features in the STS maps of g-GNRs need further clarification.

**Electronic properties of NPGs**

In Fig. 3 we proceed to characterize the electronic properties of an NPG segment comprising four fused g-GNRs. Similar to the g-GNR case, we perform d$I$/d$V$ point spectroscopy at different positions of the NPG

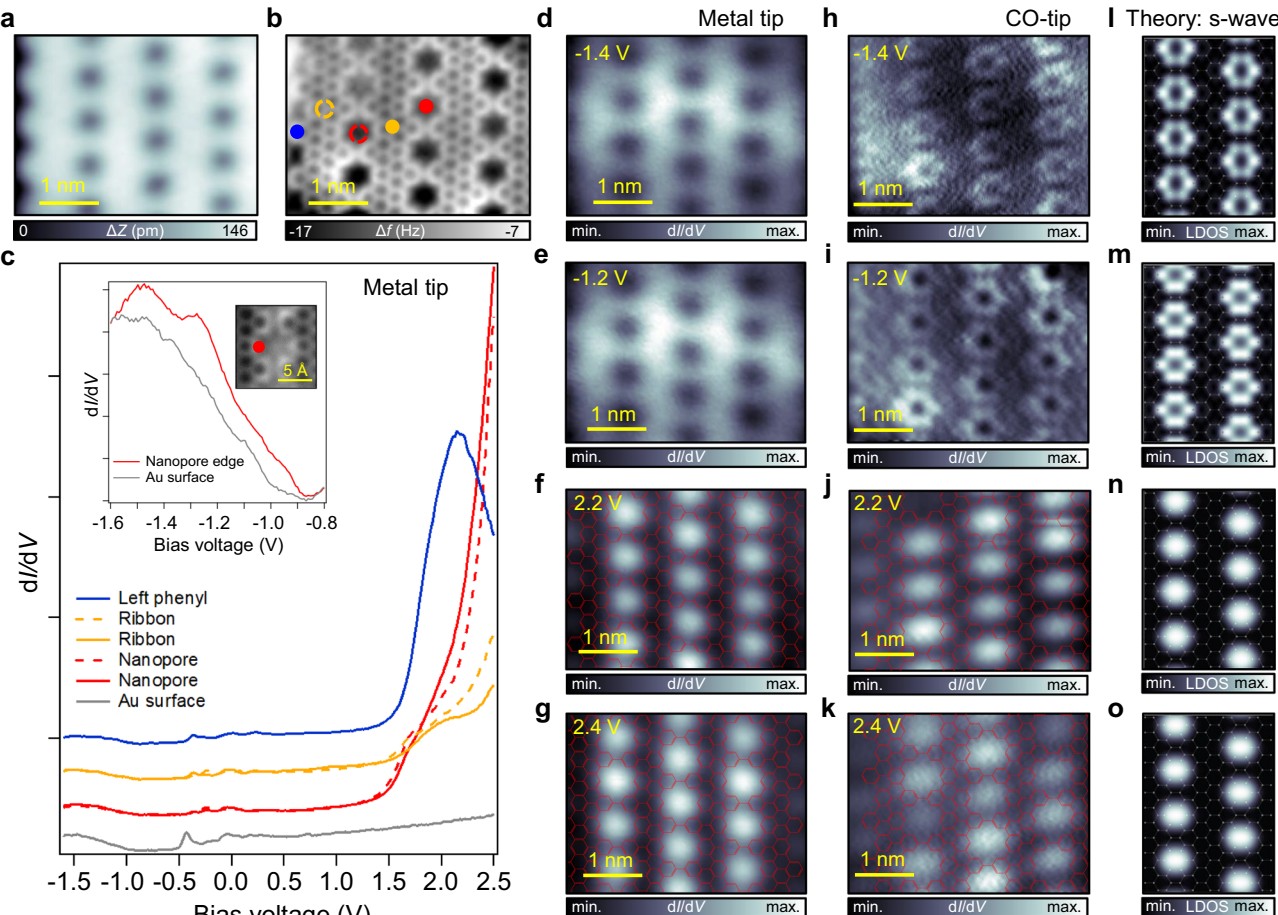

**Fig. 3 | Electronic properties of an NPG structure formed by the fusion of four g-GNRs. a**, **b** STM ($V = 100$ mV; $I = 10$ pA) and nc-AFM images of an NPG composed of four fused g-GNRs. The colored dots indicate the point spectroscopy positions. **c** d$I$/d$V$ point spectra ($I = 70$ pA; $V = -1.6$ V; $f = 954$ Hz; $V_{mod\ (0-to-peak)} = 20$ mV) acquired at different positions of the NPG [left phenyl (blue), ribbon (orange), nanopore (red)] and Au(111) substrate (gray). Despite the weak intensity of the VB onset [see STS at the inset for a clear view of the VB peak (red) as compared to Au surface (gray) ($I = 1$ nA; $V = -1.6$ V)] compared to the prominent CB (similar to the g-GNR case), a band gap of $\approx 2.7$ V can be extracted. **d**–**g** Constant height d$I$/d$V$ maps acquired with a metal tip near the VB [tip stabilization parameters: ($V = -1.4$,

$I = 200$ pA) and ($V = -1.2$ V, $I = 200$ pA)] and CB [tip stabilization parameters: ($V = 2.2$ V, $I = 200$ pA) and ($V = 2.4$ V, $I = 200$ pA)] frontier orbitals. The subtle VB features cannot be captured while the CB conductance features appear localized in the nanopores. **h**–**k** Constant height d$I$/d$V$ maps acquired with a CO-tip ($f = 353$ Hz; $V_{mod\ (0-to-peak)} = 40$ mV) close to the onsets of the VB [tip stabilization parameters: ($V = -1.4$ V, $I = 100$ pA) and ($V = -1.2$ V, $I = 200$ pA)] and CB [tip stabilization parameters: ($V = 2.2$ V, $I = 70$ pA) and ($V = 2.4$ V, $I = 70$ pA)]. **l**–**o** Corresponding DFT LDOS map simulations as obtained using appropriate energy ranges for the VBs and CBs (see Supplementary Fig. 18).

(Fig. 3a–c), namely at the nanopores (red), ribbon centers (orange), and edge phenyls (blue) using a metal tip. Again, the d$I$/d$V$ spectra for the NPG are very asymmetric, where a very intense shoulder can be seen at $\approx 1.8$ V, located mainly at the nanopore centers (red spectra). The features in the occupied region are again less pronounced, but a peak can be observed at $-1.2$ V at the NPG nanopore edge (see spectra in the inset). The bandgap ($\approx 2.7$ V) is slightly smaller than for single g-GNRs, as expected for coupled g-GNRs[35]. This shows that the wide bandgap is preserved upon NPG formation, contrasting the previous experimental assignment of a narrow bandgap of 1.14 eV for a two g-GNR fused NPG[37].

To study the spatial distribution of the LDOS, we perform d$I$/d$V$ maps close to the VB and CB onsets. For the VBs (Fig. 3d, e), no clear conductance features are observed with a metal tip. To increase the resolution of the subtle VBs features, a CO-tip is used in Fig. 3h, i. Now, the intensity appears localized at the nanopore edges, forming a hexagonal ring pattern, and at the boundaries of the NPG flake (both at $-1.4$ V and $-1.2$ V). Interestingly, the NPG edge behaves similarly to a single g-GNR edge, and the conductance features are identical to the ones observed in Fig. 2b (e.g., compare the intensity winding along the NPG edge observable in the d$I$/d$V$ map at $-1.2$ V with the one of the

g-GNR in Supplementary Fig. 4). For the unoccupied region (Fig. 3f, g, j, k), the intensity appears localized in the nanopores as well as in the gulfs at the NPG border (irrespective of whether a CO- or metal tip was used). Such nanopore conductance features have also been recently observed in holey nanographenes[3,33], molecular nanoporous networks[53], and other NPGs[35] and they were assigned differently to LUMO, NIR, IPS, or SAMO states.

Our DFT calculations of the NPG band structure confirm its semiconducting character with a slightly reduced bandgap (1.63 eV) compared to the single g-GNR (1.84 eV) (see Supplementary Fig. 17). The highly dispersive bands in both orthogonal directions of the NPG (along ΓY and across ΓX of the g-GNRs) confirm its 2D character in comparison to the more anisotropic NPGs reported so far[35,36,60]. The 2D character is also confirmed with EPWE photoemission intensity simulations where graphene-like dispersive spectral weights are predicted for orthogonal directions (see Supplementary Figs. 19 and 20).

The spatial distribution of the wave functions extracted for the VBs and CBs at the Γ point show delocalized electronic wave functions over the entire structure and the formation of bonding and anti-bonding states originating from the coupling of individual states in adjacent g-GNRs (see Supplementary Fig. 17). While the wave functions

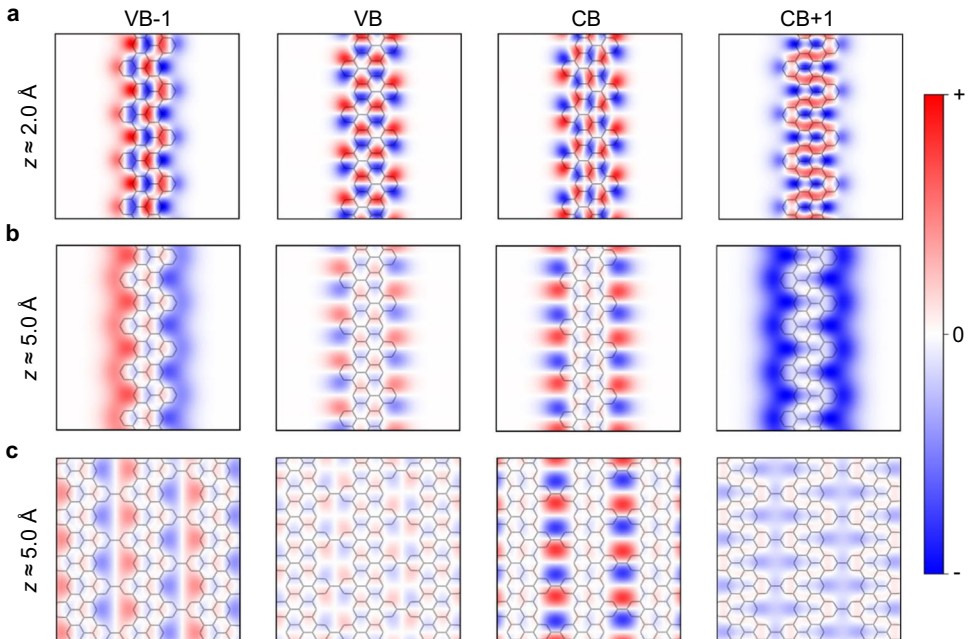

**Fig. 4 | Effect of tip-sample distance on the decay of g-GNR and NPG orbitals.** Kohn-Sham orbitals of VB-1, VB, CB, and CB + 1 at the band onsets (Γ-point) of a g-GNR (**a**, **b**) and NPG (**c**) evaluated at 2 Å (**a**) and 5 Å (**b**, **c**).

for the VBs may again resemble the experimental d$I$/d$V$ maps, the CBs look drastically different. Next, DFT-based LDOS map simulations in the Tersoff-Hamann approximation are performed at $z = 5$ Å for the VBs and CBs (Fig. 3l–o) considering a s-wave tip. The hexagonal ring and dome-shape conductance features appearing around and inside the nanopores for the VBs and CBs, respectively, are well reproduced. An energy integration of 200 meV and 250 meV is required for a satisfactory matching of occupied and unoccupied states. Such CBs nanopore features are continuously observed in the experiment for voltages as high as 3.2 V, beyond which CO-tip and NPG integrity are compromised (see Supplementary Figs. 15 and 16 for CO- and metal tips, respectively). The nanopore features are also observed for a wide energy window in DFT-based LDOS maps until the IPS is reached, which again coincidentally shows a very similar pattern to the CBs (see Supplementary Fig. 18). In the experiment the IPS, which is expected to appear beyond 3.7 V[56,57], could not be probed.

## Discussion

To disentangle the origin of the conductance features observed in STS imaging of g-GNRs and NPG, we draw upon the well-established theory of STM by Tersoff and Hamann[61]. According to it, the current measured by a s-wave tip is proportional to the LDOS at the position of the probe. Assuming that sample states are typically modeled using two-dimensional Bloch waves, their decay into the vacuum ($z$-direction) can be interpreted as a filtered Fourier expansion that, for increasing tip-sample distances, keeps only the components with vanishing lateral wave vector. This well-known filtering effect[54,62–65] can have a significant influence on the LDOS maps obtained at realistic distances from the sample, as previously demonstrated for 7-AGNRs[11].

In Fig. 4a, we follow the Kohn-Sham orbitals of a g-GNR at their band onsets evaluated at 2 Å above the carbon plane. When increasing the vertical distance to the nanoribbon, all considered orbitals exhibit significant modifications, as illustrated in Fig. 4b for a tip-sample separation of 5 Å. We find that all the explored orbitals now concentrate at the edges of the g-GNR and their amplitude is vanishing at the carbon backbone. In particular, the edge signal is more intense for VB-1 and CB + 1, as their wave functions do not change sign along the edges and, thus, they decay slower with the distance[11]. In addition, VB

and CB oscillate strongly along the edge of the nanoribbon, and they concentrate into pairs of lobes around the notched regions. These results bridge the effect of the tip-sample distance on the Kohn-Sham orbitals with the DFT-based LDOS maps shown in Fig. 2e, which we find to nicely reproduce the measured constant height d$I$/d$V$ maps in Fig. 2b. The decay of the g-GNR orbitals for different heights is summarized in Supplementary Fig. 22. This filtering procedure also reproduces the electronic orbital decay observed for 7-AGNRs[11] and nanographenes/NPGs with divacancy[26] and double void[35] types of pores, validating the generality of this effect (see Supplementary Figs. 21, 25–29).

The same tip-sample distance effect holds for the NPG structure formed by laterally fusing g-GNRs. As shown in Fig. 4c, high above the NPG, the Kohn-Sham orbitals decay faster in the carbon backbone, and their intensity becomes maximum around the outer edges of the nanopores. Particularly interesting is the increased intensity at the center of the nanopores found for the CB + 1, which give rise to LDOS signals strongly localized in the pores and resemble IPS or SAMO states (see Supplementary Fig. 23 for an evolution of the NPG orbital decay with different heights). These decay features clearly explain the characteristics of the simulated LDOS maps and are key to understand and correctly interpret the origin of the experimental STS fingerprints not only of this manuscript (Figs. 2, 3) but also of other NPG and nanographene structures previously reported[33,35]. Note that we have performed height dependent d$I$/d$V$ maps for g-GNR and NPG CBs by decreasing the tip-sample separation by ≈1 Å (see Supplementary Figs. 8 and 24). However, the conductance features always remain confined at the edges of the g-GNR and nanopores of the NPG. Measurement attempts at closer tip-sample separations already affect the integrity of the nanostructures. Thus, under these circumstances, the STM technique does not allow us to observe the transition of the conductance features into the carbon backbone.

We can conclude that the strongly localized LDOS and conductance features revealed by DFT simulations and STM experiments, respectively, arise from the combination of two factors: (1) the decay of the sample's orbitals in the vertical direction, and (2) the confinement of the orbitals into 1D and 0D regions (e.g., edges, gulfs and nanopores), as determined by the exact edge structure of the system under

consideration. While the first is a general and well-known effect reported in many surfaces and molecular assemblies[54,62–65], the second is expected to be particularly relevant in graphene-based 1D and 2D nanostructures, for many of their edge and nanopore morphologies. Note that the deceptive orbital confinement at the edges and pores explored here focuses mainly on delocalized frontier orbitals (with their wave functions stemming from the carbon backbone) and does not apply to electronic states inherently localized at the edges of graphene-based nanostructures[18,19], although the latter are also subject to a similar decay effect.

In summary, by simultaneously studying the electronic properties of g-GNRs and NPG structures, we provide an understanding of the VBs and CBs wave functions' decay into the vacuum and their confinement into the edges and nanopores as recurrently observed with STS measurements for different tip-sample separations. DFT simulations show that this deceptive effect becomes evident whenever the STM/STS methodology is used to scrutinize the electronic orbitals of carbon-based 1D and 2D nanoarchitectures with varying edge types and nanopore sizes or shapes, and is caused by a loss of Fourier components, corresponding to states of high momentum. Thus, DFT-based LDOS map simulations in the Tersoff-Hamann approximation allow us to correctly identify the experimental d$I$/d$V$ measurements with the proper corresponding VB and CB wave functions, consequently avoiding dubious interpretations. Since carbon-based nanoporous materials hold great promise as filters and sensors for biological and chemical applications, the electronic orbital decay explored here should have great implications in further understanding the interactions with guest species in typical physisorption regimes.

## Methods

### Density functional theory

DFT calculations were performed as implemented in the SIESTA code[66,67]. Core electrons were represented by norm-conserving Troullier-Martins pseudopotentials[68]. Valence electrons were described by using a double-Z polarized basis set with a 0.01 Ry energy shift, which was further extended with 3 s and 3p orbitals[69]. At the expense of computational resources, such extension allows a better characterization of IPS that could arise in vacuum regions located at edges and nanopores[35]. Exchange-correlation energies were treated in the framework of the general gradient approximation according to Perdew-Burke-Ernzerhof[70]. The Brillouin zone was sampled using Monkhorst-Pack meshes of $1 \times 51 \times 1$ and $19 \times 51 \times 1$ k-points for g-GNR and NPG, respectively[71], and a 400 Ry mesh-cutoff was employed to define the real-space grid. Vacuum regions larger than 28 Å were included in directions perpendicular to g-GNR and NPG surfaces to avoid non-physical interactions between periodic images. Atomic coordinates were optimized until forces between atoms were below 0.01 eV Å$^{-1}$ and supercells were relaxed below a pressure threshold of 0.25 GPa.

To evaluate the Kohn-Sham orbitals of g-GNRs and NPG into the tunneling gap, the corresponding wave functions can be represented by a Fourier expansion using two-dimensional Bloch waves ($\vec{k}_{\parallel}$) as follows[61]

$$\psi_{\vec{k}_{\parallel},E}(\vec{x},z) \propto \sum_{G} a_G \exp\left[i\vec{k}_G \cdot \vec{x}\right] \exp\left[-z\sqrt{\kappa^2 + \left|k_G^2\right|}\right] \quad (1)$$

Where $\vec{k}_G = \vec{G} + \vec{k}_{\parallel}$ is the sum of the 2D reciprocal lattice vector $\vec{G}$ and Bloch wave vector $\vec{k}_{\parallel} \cdot \kappa = \sqrt{2m(E_{vac}-E)}/\hbar$ is a vacuum energy ($E_{vac}$)-dependent parameter, $\vec{x}$ are the spatial coordinates parallel to the surface (the carbon plane in this case), and $z$ is the vertical distance from the surface. For large tip-sample distances, i.e., high $z$ values, only the Fourier components with vanishing lateral wave vector contribute.

Note that for the g-GNR and NPG orbitals at their band onsets (Γ-point) shown in Fig. 4, $\vec{k}_{\parallel}=0$. Besides, within the Tersoff-Hamann approximation[61] the DFT-based LDOS maps (Figs. 2e, 3l–o) can be straightforwardly obtained from the sample wave functions as described in Eq. (1).

### Experimental procedures

The DBQP precursor was synthesized according to literature procedure[37]. DBQP was sublimated from a Knudsen cell heated to ≈150 °C onto Au(111) in UHV (at background pressures around $10^{-9}$ mbar). Typical evaporation times were between 1 min and 10 min. The metal surface was cleaned by cycles of argon ion sputtering and annealing. Scanning probe measurements were performed using two set-ups. For STM, nc-AFM and STS measurements with a CO-functionalized tip, a CreaTec STM/AFM instrument in UHV conditions (≈$10^{-10}$ mbar) at a temperature of 6 K was used. STM images were recorded in constant current mode. Bond-resolved current images (BR-STM)[72] were acquired in constant height mode. For both cases, we used CO-functionalized tips[73]. Nc-AFM imaging was performed in frequency modulation mode using a qPlus sensor (resonance frequency ≈ 30 kHz, Q values of 11.000, oscillation amplitude of 80 pm). AFM scans were acquired at constant height (i.e., with the $z$-feedback switched off) at a sample bias of $V=0$ V. CO-functionalized tips provided better resolution in both topographic images and d$I$/d$V$ maps. d$I$/d$V$ measurements were recorded using a lock-in amplifier with modulation frequency and voltage ($V_{mod\ (0\text{-to-peak})}$) of 289–353 Hz and 30–40 mV, respectively. d$I$/d$V$ point spectra and maps were recorded under open feedback loop conditions. For additional STS measurements with a metal tip, a Joule-Thomson STM (Specs GmbH) in UHV conditions ($p \approx 10^{-10}$ mbar) at a temperature of 4.6 K was used. d$I$/d$V$ measurements were recorded using a lock-in amplifier with modulation frequency and voltage ($V_{mod\ (0\text{-to-peak})}$) of 954 Hz and 20 mV, respectively. STM images were subjected to standard corrections, i.e., plane-subtraction, as well as global brightness/contrast adjustments. The images were processed using WSxM[74] and SpmImage Tycoon[75].

## Data availability

The data that supports the findings of this paper are available from the corresponding authors upon request.

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

## Acknowledgements

This work was supported by grant PID2019-107338RB-C66 funded by MCIN/AEI/10.13039/501100011033, grant TED2021-132388B-C44 funded by MCIN/AEI/10.13039/501100011033 and Unión Europea Next Generation EU/PRTR, and grant PID2022-140845OB-C66 funded by MCIN/AEI/10.13039/501100011033 and FEDER Una manera de hacer Europa (A.G.-L.). A.G.-L also acknowledges the financial support received from the IKUR Strategy under the collaboration agreement between Ikerbasque Foundation and DIPC on behalf of the Department of Education of the Basque Government. A.R. acknowledges funding by the Deutsche Forschungsgemeinschaft (DFG, German Research Foundation) – 453903355. I.P.-Z is grateful to Dimas García de Oteyza and Sergio Salaverria for fruitful discussions.

## Author contributions

I.P-Z. and A.G-L. conceived this project. I.P-Z., E.C-R. and A.R. conducted STM/STS and AFM measurements. K.S. and B.Y. provided technical assistance during experiments. I.P-Z. and E.C-R. performed the STM/STS and AFM data analysis. W.A. and J.V.B. provided resources and supervision. M.A.K. and Z.M.A. conducted EPWE simulations. X.D.C. and A.G-L. performed the DFT calculations. S.N., T.K., and H.S. synthesized and purified the precursor molecule. I.P-Z., X.D.C., A.G-L., and Z.M.A. contributed to writing the manuscript. All authors contributed to the revision and final discussion of the manuscript.

## Funding

## Competing interests

The authors declare no competing interests.
