## [Peer Review File · Nature Communications]

Deceptive orbital confinement at edges and pores of carbon-based 1D and 2D nanoarchitecturesREVIEWER COMMENTS

Reviewer #1 (Remarks to the Author):

The authors study the frontier orbitals of nanoporous graphene and graphene nanoribbons on Au(111) with STM. Specifically, by performing constant height STS with a CO functionalized tip. The authors address a well-known problem in the STM community regarding the imaging of molecular orbitals: the spectroscopic resonances originated from molecular orbitals appear to be normally localized at the edges or even out of the molecule. A comprehensive theoretical analysis suggest that the observed molecular distortion has its origin in the decay of the molecular orbitals in the out of plane direction (the Kohn-sham orbitals decay faster in the carbon backbone) and that the orbitals are confined in 0D and 1D regions. Although I do agree with the main message and the theoretical interpretation is very plausible, I have a main concern with the title of the manuscript that should be clarify:

- 1) While theoretical calculations support the title "Deceptive orbital confinement at the edges and pores of carbon-based 1D and 2D nanoarchitectures" I don't find experimental proofs convincing enough. To image conduction and valence bands the authors performed constant height STS at an applied sample bias of -1.3 V, -1.1 V, 1.9 V, 2.1 V for the g-GNR and -1.2 V, -1 V, 2.2 V and 2.4 V for the nanoporous graphene. Although the current setpoint prior to open the feedback is not written anywhere, I can notice that those images are taken with the tip at a considerable distance from the sample, in other words, the predominant orbital of CO tip is probably s-wave. The fact that the tip-sample is large and thus the constant height dI/dV show predominantly contrast at the edge of the ribbon is not surprising. A convincing proof would be to image the same ribbon in a much closer tip-sample distance and observe that indeed, there is no confinement at the edge while at large tip-sample distance there is an apparent increase in the confinement at the edge. That would, in my opinion, support the title of the manuscript.
- 2) I do not observe the p-wave character of the CO tip, i.e. increased LDOS in the nodal points of the molecular orbitals. This fact reinforces the idea that possibly the tip-sample distance is large and the CO tip predominant orbital is s-wave.
- 3) Jingcheng Li et al (<https://www.nature.com/articles/s41467-021-25688-z>) (among other papers showing similar results) show confined states at the edge of the GNR while at the same time showing the geometrical structure of the GNR pointing that, in this case, the tip-sample distance is very small. I have a problem conciliating those results with the title of this manuscript. What is the criterion to claim deceptive orbital confinement?
- 4) Finally, in the conclusions it is written "Thus, DFT LDOS map simulations in the Tersoff-Hamann approximation allow us to correctly identify the experimental dI/dV measurements with the proper corresponding VB and CB wave functions avoiding dubious interpretations" Wouldn't make sense to combine the theoretical approach with closer tip-sample measurements to confirm this point?

Minor Comments:

- The number of abbreviations in the paper is overwhelming which makes the understanding of the manuscript difficult.
- in page 13 the authors define the 4-NPG as: In Figure 3 we proceed to characterize the electronic properties of a NPG segment comprising four fused g-GNRs (hereafter 4-NPG). The nomenclature changes from 4-NPG to NPG to my understand arbitrarily, it gets confusing.
- Following the rational of figure 2, top and bottom panel of Fig.3c are in reverse order.
- I cannot find anywhere in the manuscript the corresponding current setpoint (I_{set}) prior to opening the feedback for the constant height dI/dV maps.

Overall, the paper is of interest in the community since it adresses a question that has been open of decades. Additionally, it asks for caution regarding the increasing number of publications claiming to observe edge states in graphene nanoribbons and related allotropes.

I find this work relevant and worth for publication in Nature Communications, however the above

mentioned comments should be satisfactorily adressed before publishing.

Reviewer #2 (Remarks to the Author):

See attached file for comments.

Review report on paper titled ‘Deceptive orbital confinement at the edges and pores of carbon-based 1D and 2D nanoarchitectures’ by Ignacio Piquero-Zulaica *et al.*

The manuscript reports on a scanning tunneling microscopy/spectroscopy (STM/STS) study on graphene nanoribbons and nanoporous graphene. The key result of the study is to identify an artefact that results in a distortion of the electronic orbitals observed in STM experiments on nanostructured graphene. In many STM studies (widely cited in the manuscript) on graphene nanoribbons (GNRs), nanographenes, and similar other graphene-based materials features of high conductance are observed at the edges of the materials. Such observations have in the past been discussed in terms of different edge-localized states. If these previous studies have been influenced by an artefact, addressing the issue would be a noteworthy advancement of the field.

The study is well made, the data analysis is robust, and the results support the conclusions. Briefly, the authors synthesize gulf-type GNRs (g-GNRs) using on-surface synthesis following well-established methods. Upon further annealing, the g-GNRs merge into larger structures among which nanoporous graphene can be identified. The graphene structures are investigated using atomic force microscopy (AFM) using a CO-functionalized tip and STM. The combined microscopy approach allows imaging the structures with bond-resolved resolution and probing the local density of states. The authors observe the edge-localized conductance features at the core of the study. The results are compared to density-functional theory (DFT) calculated orbitals. It is observed that the conductance features do not well resemble the calculated orbitals. The key result of the study is to observe from the DFT calculations that due to decay of the orbitals in the direction away from the sample plane, the conductance pattern can be significantly different from the orbitals at the graphene nanostructure. Essentially orbitals lose the high-resolution features upon moving away from the sample plane.

Beyond a few minor suggestions, I have no criticism to the study. It is of high quality. My main concern is the importance and impact of the findings to the field. As discussed in the manuscript, in Ref. 11 of the manuscript the observed effect has already been (partly) discussed. I therefore find the present contribution a good reminder and contribution to researchers working on STM of nanographenes. However, I am somewhat hesitant if it

will capture broad interest of the readership of nature communications. The authors must more convincingly argue the impact and novelty of their work.

Technical remarks:

1. The discussion around Fig. 2 is somewhat confusing. First the orbitals in Fig. 2e are discussed and then the discussion jumps back to earlier panels.
2. The dashed circles in the inset of Fig. 2a are poorly visible.

Reviewer #3 (Remarks to the Author):

Piquero-Zulaica et al. used Au(111)-supported semiconducting gulf-type GNRs (g-GNRs) and NPGs as model systems to study the electronic properties of g-GNRs and NPG structures. Due to the distortion effect that masks the undisturbed electronic orbital morphologies, the electronic states of g-GNRs and NPG nanostructures is confined along the edges and nanopores, respectively. DFT calculations confirmed that these electronic states originate from delocalized valence and conduction bands. The apparent electronic orbital distortion and confinement observed were caused by a loss of Fourier components, and this loss is related to the distance between the tip and the sample. This study is technically very well executed, and DFT LDOS map simulations in the Tersoff-Hamann approximation used in the manuscript allow people to correctly identify the experimental dI/dV measurements with the proper corresponding VB and CB wave functions avoiding dubious interpretations.

Major issues:

1. The manuscript points out that the electronic orbital distortion and confinement observed were caused by a loss of Fourier components, and the loss is related to the distance between the tip and the sample. Thus, the constant-height dI/dV maps at different tip-sample distances of g-GNRs and NPG nanostructures are necessary, which is helpful to observe various electronic states caused by different tip-sample distances in real space.

2. The g-GNRs and NPG nanostructures synthesized in the manuscript have been reported (Chem. Asian J. 10.1002/asia.201901328). Therefore, the main driving force for this work that may publish in Nature Communications is the electronic orbital distortion of two carbon-base nanostructures. However, the resolution of dI/dV spectra of the NPG nanostructure is very low, especially in the negative bias region. The description of "The shoulder in the occupied region is again less pronounced, but it can be observed at -1.0 eV at the NPG edge (green spectrum)." is problematic. Because the peak of -1.0 eV also appears on the characteristic spectra of Au(111). More importantly, the experimental dI/dV map at -1.0 eV is not consistent with the DFT simulated map at the VB. Authors should re-verify their dI/dV spectra and redefine the VBM and CBM of NPG nanostructure to avoid misleading readers.

Other issues:

3. In Figures S2 and S3, in addition to the p-GNRs, some other carbon nanostructures were also presented. What are they? Could the authors provide more information on these structures?

4. I was surprised to find that an article published in Chem. Asian J. in 2019 (Chem. Asian J. 10.1002/asia.201901328), whose corresponding author is the same person as this article. In both articles, g-GNRs and NPG nanostructures were synthesized, but the definition of the bandgaps of g-GNRs (2.7 eV vs 1.1 eV) and NPG nanostructure (2.6 eV vs 1.14 eV) is completely different. I hope the authors can make a reasonable explanation.

Reviewer #4 (Remarks to the Author):

Report "Deceptive orbital confinement at the edges and pores of carbon-based 1D and 2D nanoarchitectures" by I. Piquero-Zulaica et al.

The author report on the synthesis and characterization of quasi-1D and 2D graphene derived nanostructures (GDN) by SPM/STS in conjunction with DFT simulations. As molecular precursors the authors use DBQP which has been shown to form covalent extended quasi-1D (notched edge) and 2D (nanopore) structure of high quality on Au(111) before. The electronic structure of the GND are investigated by point dI/dV spectroscopy and dI/dV mapping. The authors identify the VB-1, VB band edge around -1.2 eV and the CB and CB+1 around +2.0 eV. They emphasize the importance of quantitatively correct STM simulations by DFT by taking account of the wave function cancelation in certain regions at larger tip-sample distances.

The data presented by the authors is of high quality and the interpretation is sound and well supported by experiments and simulation. However, the novelty, pertinence and originality of the work is not on a level to be of broader interest to the readers of Nat. Commun. The synthesis of the structures has been published before by other authors and also the importance of simulating dI/dV maps at the

appropriate height in order to take into account the effects of parity cancelation in orbital mapping is well established. In this regard, this work doesn't add to the experimental/theoretical works of Söde et al. or the theoretical work by Tersoff et al. for the wave function extrapolation. The novelty of this paper is the specific spectroscopic characterization of the frontier orbitals of the DBQP ribbons and fused ribbons on a level routinely done today for nanocarbon materials. Therefore, I would propose a more specialized journal like Carbon or Phys. Condens. Matter.

Specific remarks

Page 3: I do not understand why the authors claim that the explanation of Söde et al. is 'tentative', it is supported by experiment and simulation in real and reciprocal space.

Page 3: It is not clear what a 'wave function morphology' is and also not what this distortion would be. The wave function $\psi(x,y,z)$ is mapped in STS as $|\psi(x,y,z(x,y))|^2$ to correctly simulate the dI/dV map it needs to be evaluated along the $z(x,y)$ contours, but that is not a distortion.

It is not clear to me if the dI/dV maps are measured with a CO-functionalized tip, in that case the simulation need to be conducted with a p-wave (or s- and p-wave mixture) tip wave function. The simulations are s-wave type so this could be at odds with a CO-tip. This possible ambiguity should be addressed.

Reviewer 1

The authors study the frontier orbitals of nanoporous graphene and graphene nanoribbons on Au(111) with STM. Specifically, by performing constant height STS with a CO functionalized tip. The authors address a well-known problem in the STM community regarding the imaging of molecular orbitals: the spectroscopic resonances originated from molecular orbitals appear to be normally localized at the edges or even out of the molecule. A comprehensive theoretical analysis suggest that the observed molecular distortion has its origin in the decay of the molecular orbitals in the out of plane direction (the Kohn-sham orbitals decay faster in the carbon backbone) and that the orbitals are confined in 0D and 1D regions. Although I do agree with the main message and the theoretical interpretation is very plausible, I have a main concern with the title of the manuscript that should be clarify:

We thank the reviewer for the positive assessment of our manuscript.

1) While theoretical calculations support the title “Deceptive orbital confinement at the edges and pores of carbon-based 1D and 2D nanoarchitectures” I don’t find experimental proofs convincing enough. To image conduction and valence bands the authors performed constant height STS at an applied sample bias of -1.3 V, -1.1 V, 1.9 V, 2.1 V for the g-GNR and -1.2 V, -1 V, 2.2 V and 2.4 V for the nanoporous graphene. Although the current setpoint prior to open the feedback is not written anywhere, I can notice that those images are taken with the tip at a considerable distance from the sample, in other words, the predominant orbital of CO tip is probably s-wave. The fact that the tip-sample is large and thus the constant height dI/dV show predominantly contrast at the edge of the ribbon is not surprising. A convincing proof would be to image the same ribbon in a much closer tip-sample distance and observe that indeed, there is no confinement at the edge while at large tip-sample distance there is an apparent increase in the confinement at the edge. That would, in my opinion, support the title of the manuscript.

Regarding the current set-point prior to opening the feedback, the reviewer is right and this information was not provided in the initial version. Now we include all the current set-points either in the figure panels or captions. For the CO functionalized tip measurements, indeed we used set-point currents that are relatively low (<200 pA in our CreaTec STM-AFM set-up). Therefore, as pointed out by the reviewer, the

predominant orbital of the CO tip is s-wave. This leads to a satisfactory matching of the experimental dI/dV maps with DFT simulations performed considering a s-wave tip.

For completeness, we have performed additional STS measurements in a different machine (Joule-Thomson STM operated at 4.6 K) by using a metal tip. We have also performed dI/dV maps (at several voltages mainly in the CB region) for decreasing tip-sample distances (by changing the current set-point from 100 pA to 1 nA) for g-GNR and NPG. However, the conductance features remain unchanged (most clearly observed for the CBs in the unoccupied region, see Figure R1 and Supplementary Figs. 24 and 8). This is not surprising since in STM an order of magnitude change in the set-point current only approaches the tip by $\sim 1\text{\AA}$. These new measurements performed clearly support the fact that in STM, the measurement regime (i.e., the tip-sample distance) is qualitatively comparable to the DFT simulations performed with an s-wave tip in the range of 5 \AA away from the nanostructures (see Fig. 4 in the main). Therefore, the experimental observation of the non-confined conductance features (qualitatively corresponding to the 2 \AA range in DFT calculations) is challenging and not achievable in our STM set-ups. Note that dI/dV maps performed at closer tip-sample separations would affect the integrity of the g-GNRs and NPGs.

Figure R1: Constant-height dI/dV maps acquired at the g-GNR and NPG CBs with a metal tip. The current set-point is changed from 100 pA to 1 nA, which makes the STM tip to approach to the g-GNR and NPG by $\sim 1\text{\AA}$.

We have introduced the following text in the main manuscript (page 17): *“Note that we have performed height dependent dI/dV maps for g-GNR and NPG CBs by decreasing the tip-sample separation by $\sim 1\text{\AA}$ (see Supplementary Figs. 8 and 24). However, the conductance features always remain confined at the edges of the g-GNR and nanopores of the NPG. Measurement attempts at closer tip-sample separations already affect the integrity of the nanostructures. Thus, under these circumstances, the*

STM technique does not allow us to observe the transition of the conductance features into the carbon backbone.

The title of our manuscript clearly states that the observation of STS conductance features at the edges of g-GNRs and nanopores in NPGs is deceptive because they tend not to directly match with the frontier orbital wave functions simulated in DFT (without considering the decay into the vacuum) and can be easily misinterpreted by attributing these conductance features to high-in-energy image potentials states (IPS) or super atom molecular orbitals (SAMO), among others [see Fig. R2 and Refs. 33,35]. In these two papers, dI/dV maps obtained in the unoccupied region (positive voltages) also showed clear conductance features confined at the nanopores (voids and double voids). They were attributed to stem from high-energy SAMO (panel (a)) and IPS (panel (b)) states since a good matching with their wave functions was found. However, taking the electronic orbital decay into account, now we have a deeper understanding on how to interpret these conductance features and can therefore assign the experimental dI/dV maps of Fig. R2 to lowest unoccupied molecular orbitals (LUMOs) and conduction bands (CBs) [see Fig. R5 for the electronic orbital decay in double-void and divacancy nanopores, and Supplementary Figs. 25 to 29]. Therefore, the term used in our title “deceptive electronic orbital decay” is appropriate.

Figure R2: Electronic orbital decay and confinement into a nanographene’s void and into an NPG’s double-voids. Figure has been adapted from Refs. 33,35. Panel a is reprinted (adapted) with permission from Hieulle et al., Nano Lett. 18, 418-423 (2018). Copyright 2018 American Chemical

Society. Panel b is reprinted (adapted) from Moreno et al., Science 360, 199-203 (2018). Reprinted with permission from AAAS.

We have strengthened this point in the manuscript (page 17). *“These decay features clearly explain the characteristics of the simulated LDOS maps and are key to understand and correctly interpret the origin of the experimental STS fingerprints not only of this manuscript (Figures 2 and 3) but also of other NPG and nanographene structures previously reported^{33,35}”*

2) I do not observe the p-wave character of the CO tip, i.e. increased LDOS in the nodal points of the molecular orbitals. This fact reinforces the idea that possibly the tip-sample distance is large and the CO tip predominant orbital is s-wave.

The reviewer is right. The maximum current set-point used for CO tip measurements is around 200 pA (indicated now in the figure captions of Figure 2 and Figure 3 of the main manuscript and figure panels in the Supplementary Information). Therefore the CO tip predominant orbital is s-wave. As mentioned above, we have performed additional STS point spectra measurements with a metal tip (Fig. R6, Supplementary Fig. 8 for the g-GNRs and Figures 3 in the main and Supplementary Fig. 16 for the NPGs), as well as dI/dV maps (at different voltages) for varying current set-points from 100 pA to 1 nA (Fig. R1 and Supplementary Figs. 24 and 8). As can be clearly seen in the g-GNR and NPG CBs (compare Figure 2 with Supplementary Fig. 8 for the g-GNRs and Figure 3 in the main for the NPG case), the dI/dV maps are identical for both tips, corroborating the s-wave character of the CO tip. Therefore the matching with DFT LDOS simulations considering a s-wave tip is satisfactory.

3) Jingcheng Li et al (<https://www.nature.com/articles/s41467-021-25688-z>) (among other papers showing similar results) show confined states at the edge of the GNR while at the same time showing the geometrical structure of the GNR pointing that, in this case, the tip-sample distance is very small. I have a problem conciliating those results with the title of this manuscript. What is the criterion to claim deceptive orbital confinement?

GNRs with edge states should have the wave function inherently located at the edge. Therefore, these edge states are not fictitious and should be observed confined at the edges in dI/dV maps. This is indeed the case for the 3,1,6-chGNR and 3,1,8-chGNRs studied by Jingcheng Li and coworkers (see Figure R3). Please note that edge states were measured in dI/dV maps at 2 mV (with a CO tip). This small voltage allows the

STM tip to approach the GNR very close, so that even bond resolved imaging could be achieved in the same image. In our case, the CBs are located at 2 V and geometrical features of the g-GNR (or NPG) at this voltage are impossible to achieve.

Figure R3: Electronic properties of topological chiral-GNRs. Figure has been adapted from Li et al. Nat Commun. 12, 5538 (2021). This is included as Ref. 18 in the manuscript references.

Another example is the observation of zigzag end states in 7-AGNRs on TiO₂(011)-(2×1). Here the spin polarized end-state wave functions were indeed observed at the edge of the GNRs (with zigzag type edge) in STS measurements (Fig. R4). As can be observed in panels (a,b) clear conductance peaks observed at the end of the GNR (red spectrum) are not present at the center of the GNR (blue spectrum). They correspond to the HOMO and LUMO wave functions confined at the end of the GNR [panel (c)].

Figure R4: Observation of zigzag end states in substrate decoupled finite 7-AGNRs. Figure has been adapted from Ref. 19 (now included in the manuscript references). Figure is reprinted (adapted) from Kolmer et al., Science 369, 571-575 (2020). Reprinted with permission from AAAS.

In our case, we focus on the frontier bands of GNRs and NPGs, namely, valence bands (VBs) and conduction bands (CBs). As observed from DFT, the wave functions

corresponding to these bands evaluated at the Γ point appear located at the carbon backbone and not at the edge only (see Supplementary Figs. 9 and 17). In other words, they are delocalized states, in contrast to intrinsically localized low energy states appearing at the edges of zigzag GNRs and chiral GNRs, or at the ends of some armchair GNRs. However, due to the wave function decay in the z-direction (away from the GNR or NPG), the experimental dI/dV maps capture the confinement of the conductance features into the edges of GNRs and nanopores in NPGs (clearly outside the carbon backbone). These features can be easily misinterpreted by comparing them with wave functions stemming from high-energy electronic states such as super-atom molecular orbitals (SAMO) [Fig. R2 (a)] or image potential states (IPS) [Figure R2(b)] (see Refs. 33,35). As can be observed, the conductance features highly resemble the calculated IPS or SAMO wave functions, leading to a misinterpretation. Therefore the title of our manuscript containing the word “deceptive” is justified.

A new sentence regarding edge states is added in the introduction (page 4): *“It should be noted that for instance the edge states of topological chiral-GNRs¹⁸ and the zigzag end states of substrate-decoupled finite 7-AGNRs¹⁹ are inherently localized and therefore experimentally observed at the edges as well.”* Also at the end of the discussion section (page 18) we add: *“Note that the deceptive orbital confinement at the edges and pores explored here focuses mainly on delocalized frontier orbitals (with their wave functions stemming from the carbon backbone) and does not apply to electronic states inherently localized at the edges of graphene-based nanostructures^{18,19}, although the latter are also subject to a similar decay effect”.*

A sentence highlighting the deceptive character of the electronic orbitals is included in the manuscript (page 17). *“These decay features clearly explain the characteristics of the simulated LDOS maps and are key to understand and correctly interpret the origin of the experimental STS fingerprints not only of this manuscript (Figures 2 and 3) but also of other NPG and nanographene structures previously reported^{33,35}”.*

4) Finally, in the conclusions it is written “Thus, DFT LDOS map simulations in the Tersoff-Hamann approximation allow us to correctly identify the experimental dI/dV measurements with the proper corresponding VB and CB wave functions avoiding dubious interpretations” Wouldn't make sense to combine the theoretical approach with closer tip-sample measurements to confirm this point?

Following the reviewer's suggestion, we have measured constant-height dI/dV maps for the g-GNR and NPG CBs with increasing current set-point (from 100 pA to 1 nA), allowing us to approach the tip towards the sample by ~ 1 Å. However, the conductance features remain confined at the edges of the g-GNRs and NPG nanopores. The results can be observed in Fig. R1 and Supplementary Figs. 8 and 24. Please refer to point 1 for more details regarding the implications of our new observations.

Minor Comments:

- The number of abbreviations in the paper is overwhelming which makes the understanding of the manuscript difficult.

We have tried to reduce the abbreviations as much as possible now (see next point). However, we consider that abbreviations such as STM, STS, nc-AFM, GNR, NPG, HOMO, LUMO, IPS, NIR, PIR and SAMO are widely used and should be kept in the manuscript.

- in page 13 the authors define the 4-NPG as: In Figure 3 we proceed to characterize the electronic properties of a NPG segment comprising four fused g-GNRs (hereafter 4-NPG). The nomenclature changes from 4-NPG to NPG to my understand arbitrarily, it gets confusing.

From now on, we only use the NPG abbreviation for all the cases.

- Following the rational of figure 2, top and bottom panel of Fig.3c are in reverse order.

Figure 3 has been changed accordingly.

- I cannot find anywhere in the manuscript the corresponding current setpoint (I_{set}) prior to opening the feedback for the constant height dI/dV maps.

The reviewer is right. In the new version, all the current set-points have been included. They appear in the figure panels or captions.

Overall, the paper is of interest in the community since it adresses a question that has been open of decades. Additionally, it asks for caution regarding the increasing number of publications claiming to observe edge states in graphene nanoribbons and related allotropes.

I find this work relevant and worth for publication in Nature Communications, however the above mentioned comments should be satisfactorily adressed before publishing.

We thank the reviewer for the positive assessment of the quality and importance of this study.

Reviewer 2

The manuscript reports on a scanning tunneling microscopy/spectroscopy (STM/STS) study on graphene nanoribbons and nanoporous graphene. The key result of the study is to identify an artefact that results in a distortion of the electronic orbitals observed in STM experiments on nanostructured graphene. In many STM studies (widely cited in the manuscript) on graphene nanoribbons (GNRs), nanographenes, and similar other graphenebased materials features of high conductance are observed at the edges of the materials. Such observations have in the past been discussed in terms of different edge-localized states. If these previous studies have been influenced by an artefact, addressing the issue would be a noteworthy advancement of the field.

We thank the reviewer for considering our work a noteworthy advancement of the field.

The study is well made, the data analysis is robust, and the results support the conclusions. Briefly, the authors synthesize gulf-type GNRs (g- GNRs) using on-surface synthesis following well-established methods. Upon further annealing, the g- GNRs merge into larger structures among which nanoporous graphene can be identified. The graphene structures are investigated using atomic force microscopy (AFM) using a CO-functionalized tip and STM. The combined microscopy approach allows imaging the structures with bond-resolved resolution and probing the local density of states. The authors observe the edge-localized conductance features at the core of the study. The results are compared to density-functional theory (DFT) calculated orbitals. It is observed that the conductance features do not well resemble the calculated orbitals. The key result of the study is to observe from the DFT calculations that due to decay of the orbitals in the direction away from the sample plane, the conductance pattern can be significantly different from the orbitals at the graphene nanostructure. Essentially orbitals lose the high-resolution features upon moving away from the sample plane.

Beyond a few minor suggestions, I have no criticism to the study. It is of high quality.

We thank the reviewer for such a positive response to our work.

My main concern is the importance and impact of the findings to the field. As discussed in the manuscript, in Ref. 11 of the manuscript the observed effect has already been

(partly) discussed. I therefore find the present contribution a good reminder and contribution to researchers working on STM of nanographenes. However, I am somewhat hesitant if it will capture broad interest of the readership of nature communications. The authors must more convincingly argue the impact and novelty of their work.

In this work we provide a deep understanding of the frontier orbitals decay into the vacuum and their confinement into the edges and nanopores of any 1D or 2D graphene-based nanostructure (e.g., GNRs, nanographenes and NPGs). To generalize this effect (strengthening the impact and novelty of our work), we have performed additional DFT calculations for other two types of nanopores: divacancies and double-voids (see Fig. R5). Please note that GNRs with divacancies and NPGs with double-voids have already been synthesized on surfaces, therefore they are very relevant [Ref. 26, 35]. We find that the wave function decay into the vacuum and its confinement into the nanopores is a general effect, regardless of the pore size and shape (Figure R5 and Supplementary Figs. 25 to 29).

Figure R5: Effect of tip-sample distance on the electronic orbital decay of GNRs with periodically spaced double-voids and divacancies. Kohn-Sham orbitals of CB+1 at the band onsets (Γ -point) of a GNR evaluated from 2Å to 6Å.

These additional DFT calculations support the fact that dubious interpretations may have been given to the conductance features observed on previous nanographene and NPG studies [see Figure R2 and Refs. 33,35]. In these two papers, dI/dV maps obtained in the unoccupied region (positive voltages) also showed clear conductance

features confined at the nanopores (voids and double voids). They were attributed to stem from high-energy SAMO (panel (a)) and IPS (panel (b)) states since a good matching with their wave functions was found. However, taking the electronic orbital decay into account, now we have a deeper understanding on how to interpret these conductance features and can therefore assign the experimental dI/dV maps of Fig. R2 to lowest unoccupied molecular orbitals (LUMOs) and conduction bands (CBs) [see Fig. R5 for the electronic orbital decay in double-void and divacancy nanopores, and Supplementary Figs. 25 to 29].

We have added the following paragraphs to convincingly argue the impact and novelty of our work (page 16-17): "*This filtering procedure also reproduces the electronic orbital decay observed for 7-AGNRs¹¹ and nanographenes/NPGs with divacancy²⁶ and double void³⁵ types of pores, validating the generality of this effect (see Supplementary Figs. 21, 25-29).*"

And the following sentence on page 17: "*These decay features clearly explain the characteristics of the simulated LDOS maps and are key to understand and correctly interpret the origin of the experimental STS fingerprints not only of this manuscript (Figures 2 and 3) but also of other NPG and nanographene structures previously reported^{33,35}*"

Technical remarks:

1. The discussion around Fig. 2 is somewhat confusing. First the orbitals in Fig. 2e are discussed and then the discussion jumps back to earlier panels.

We have changed the order of the panels.

2. The dashed circles in the inset of Fig. 2a are poorly visible.

The circles have been enlarged.

Reviewer 3:

Piquero-Zulaica et al. used Au(111)-supported semiconducting gulf-type GNRs (g-GNRs) and NPGs as model systems to study the electronic properties of g-GNRs and NPG structures. Due to the distortion effect that masks the undisturbed electronic orbital morphologies, the electronic states of g-GNRs and NPG nanostructures is confined along the edges and nanopores, respectively. DFT calculations confirmed that these electronic states originate from delocalized valence and conduction bands. The apparent electronic orbital distortion and confinement observed were caused by a loss of Fourier components, and this loss is related to the distance between the tip and the sample. This study is technically very well executed, and DFT LDOS map simulations in the Tersoff-Hamann approximation used in the manuscript allow people to correctly identify the experimental dI/dV measurements with the proper corresponding VB and CB wave functions avoiding dubious interpretations.

We thank the reviewer for appreciating our work.

1.The manuscript points out that the electronic orbital distortion and confinement observed were caused by a loss of Fourier components, and the loss is related to the distance between the tip and the sample. Thus, the constant-height dI/dV maps at different tip-sample distances of g-GNRs and NPG nanostructures are necessary, which is helpful to observe various electronic states caused by different tip-sample distances in real space.

The reviewer is right. We have performed \$dI/dV\$ maps for different voltages and for each voltage the current set-point is varied (from 100 pA to 1 nA) so that the tip can approach the g-GNRs and NPGs by \$\sim 1 \text{ \AA}\$. These measurements have been performed with a metal tip and they complement the previous STS performed with a CO-tip. These new results are summarized in Fig. R1 and in Supplementary Figs. 8 and 24.

As can be observed in Fig. R1, the conductance features of the CBs remain unchanged (also see Supplementary Figs. 24 and 8). This is not surprising since in STM an order of magnitude change in the set-point current only approaches the tip by \$\sim 1 \text{ \AA}\$. These new measurements performed clearly support the fact that in STM, the measurement regime (i.e., the tip-sample distance) is qualitatively comparable to the DFT simulations performed with an s-wave tip in the range of \$5 \text{ \AA}\$ away from the nanostructures (see Fig. 4 in the main). Therefore, the experimental observation of the non-confined conductance features (qualitatively corresponding to the \$2 \text{ \AA}\$ range in DFT calculations)

is challenging and not achievable in our STM set-ups. Note that dl/dV maps performed at even closer tip-sample separations would affect the integrity of the g-GNRs and NPGs.

We have introduced the following text in the main manuscript (page 17): *“Note that we have performed height dependent dl/dV maps for g-GNR and NPG CBs by decreasing the tip-sample separation by ~ 1 Å (see Supplementary Figs. 8 and 24). However, the conductance features always remain confined at the edges of the g-GNR and nanopores of the NPG. Measurement attempts at closer tip-sample separations already affect the integrity of the nanostructures. Thus, under these circumstances, the STM technique does not allow us to observe the transition of the conductance features into the carbon backbone.”*

2. The g-GNRs and NPG nanostructures synthesized in the manuscript have been reported (Chem. Asian J. 10.1002/asia.201901328). Therefore, the main driving force for this work that may publish in Nature Communications is the electronic orbital distortion of two carbon-base nanostructures. However, the resolution of dl/dV spectra of the NPG nanostructure is very low, especially in the negative bias region. The description of “The shoulder in the occupied region is again less pronounced, but it can be observed at -1.0 eV at the NPG edge (green spectrum).” is problematic. Because the peak of -1.0 eV also appears on the characteristic spectra of Au(111). More importantly, the experimental dl/dV map at -1.0 eV is not consistent with the DFT simulated map at the VB. Authors should re-verify their dl/dV spectra and redefine the VBM and CBM of NPG nanostructure to avoid misleading readers.

The reviewer is right. The synthesis was reported before (Ref. 37), but the atomic scale characterization with nc-AFM, as well as high-quality STS measurements (at different positions of the g-GNRs and NPGs and dl/dV maps) were not reported.

Regarding the verification of the NPG bandgap, we agree with the reviewer. Therefore, we have performed new STS measurements with a metal tip and we identify the NPG onset of the VB around -1.2 V and the onset of the CB at around 1.6 V, inducing a bandgap of ≈ 2.7 V (see Fig. R6, Figure 3 in the main and dl/dV maps in Supplementary Figs. 16 and 24). Now the NPG bandgap determination is rigorous and the dl/dV map at -1.2 V is consistent with the DFT simulated map for the onset of the CB.

For completeness, we have also verified the band gap of the g-GNR using a metal tip and the same value as with the CO tip ($\approx 2.8\text{V}$) is obtained (see Fig. R6 and compare Fig. 2 of the main with Supplementary Fig. 8).

Figure R6: g-GNR and NPG STS bandgap measurements with a metal tip. (a) STS acquired with a metal tip on the g-GNR where the onset of the VB is located around -1.2 V and the one of the CB at 1.8 V . (b,c) dI/dV maps (CO-tip) corroborating the onset of the VB and CB. (d) STS acquired with a metal tip on the NPG where the onset of the VB is located around -1.2 V and the onset of the CB at 1.6 V . (e,f) dI/dV maps (with CO and metal tip, respectively) corroborating the onset of the VB and CB.

Other issues:

3. In Figures S2 and S3, in addition to the p-GNRs, some other carbon nanostructures were also presented. What are they? Could the authors provide more information on these structures?

In Figure R7 and Supplementary Fig. 3, nc-AFM images of the most common defects are summarized. The most common defect is related to the formation of triphenylene molecules embedded in between g-GNR segments. We suggest that these defects happen during the g-GNR planarization process (already below 500°C). It is known that above 400°C , segment flipping and carbon atom rearrangement effects can take place. During the segment flipping, a phenyl ring may be detached from the g-GNR, giving rise to terphenyl defects (see Fig. R7). Apart from the terphenyl defects, other types of nanographene type of defects can also be observed (see Supplementary Fig. 3). We have introduced a discussion paragraph on page 3 of the Supplementary Information. Please note that a more rigorous study of the defect types and their formation mechanisms is beyond the scope of this paper.

Figure R7: Identification of a triphenylene defect in g-GNRs with nc-AFM.

4. I was surprised to find that an article published in Chem. Asian J. in 2019 (Chem. Asian J. 10.1002/asia.201901328), whose corresponding author is the same person as this article. In both articles, g-GNRs and NPG nanostructures were synthesized, but the definition of the bandgaps of g-GNRs (2.7 eV vs 1.1 eV) and NPG nanostructure (2.6 eV vs 1.14 eV) is completely different. I hope the authors can make a reasonable explanation.

The reviewer is right on the fact that the on-surface synthesis was first reported in Ref. 37. Regarding the bandgap of the g-GNR and NPG, in our STS measurements we clearly observe larger bandgaps than the ones reported before. In the initial version of the manuscript, STS was performed with a CO tip. As pointed out by reviewer 3 in point 2, the onset of the VB in the NPG case was not unambiguously determined in our previous version. Therefore, now we have verified the bandgaps of g-GNRs and NPG with a metal tip (see Fig. R6, Figure 3 of the main and Supplementary Figs. 8 and 16). These measurements have been performed in a second low temperature STM machine (JT-STM operated at 4.6 K, see methods). We can conclude that the bandgaps for g-GNR and NPG are ≈ 2.8 V and ≈ 2.7 V, respectively. These values have been corrected in the manuscript (pages 10 and 13).

We can only speculate on the reason why the previous bandgap report is incorrect, which, most probably, can happen due to tip-effects, common in high-coverage samples where tip preparation by indentation becomes complex.

Note that Prof. Sakaguchi provided us with the DBQP precursor in this work. The experimental bandgap measurements were performed by I.P-Z, E.C-R. and A.R. (see Author Contributions section on page 21).

Reviewer 4:

The author report on the synthesis and characterization of quasi-1D and 2D graphene derived nanostructures (GDN) by SPM/STS in conjunction with DFT simulations. As molecular precursors the authors use DBQP which has been shown to form covalent extended quasi-1D (notched edge) and 2D (nanopore) structure of high quality on Au(111) before. The electronic structure of the GND are investigated by point dl/dV spectroscopy and dl/dV mapping. The authors identify the VB-1, VB band edge around -1.2 eV and the CB and CB+1 around +2.0 eV. They emphasize the importance of quantitatively correct STM simulations by DFT by taking account of the wave function cancelation in certain regions at larger tip-sample distances. The data presented by the authors is of high quality and the interpretation is sound and well supported by experiments and simulation. However, the novelty, pertinence and originality of the work is not on a level to be of broader interest to the readers of Nat. Commun. The synthesis of the structures has been published before by other authors and also the importance of simulating dl/dV maps at the appropriate height in order to take into account the effects of parity cancelation in orbital mapping is well established. In this regard, this work doesn't add to the experimental/theoretical works of Söde et al. or the theoretical work by Tersoff et al. for the wave function extrapolation. The novelty of this paper is the specific spectroscopic characterization of the frontier orbitals of the DBQP ribbons and fused ribbons on a level routinely done today for nanocarbon materials. Therefore, I would propose a more specialized journal like Carbon or Phys. Condens. Matter.

We agree with the reviewer that the synthesis of g-GNRs and NPGs has been reported in Ref. 37. However, we are certain that our work represents a clear step forward in the field (as clearly stated by Reviewer 2), providing topographic characterization at the atomic scale (STM and nc-AFM), electronic property studies (including the correction of the bandgap reported before) and most importantly, a deep understanding of the electronic orbital decay and confinement into the edges of GNRs and nanopores in NPGs (of general character and broad interest to the scientific community).

In this work we provide a deep understanding of the frontier orbitals decay into the vacuum and their confinement into the edges and nanopores of any 1D or 2D graphene-based nanostructure (e.g., GNRs, nanographenes and NPGs). To

generalize this effect (strengthening the impact and novelty of our work), we have performed additional DFT calculations for other two types of nanopores: divacancies and double-voids (see Fig. R5). Please note that GNRs with divacancies and NPGs with double-voids have already been synthesized on surfaces, therefore they are very relevant [Ref. 26, 35]. We find that the wave function decay into the vacuum and its confinement into the nanopores is a general effect, regardless of the pore size and shape (Figure R5 and Supplementary Figs. 25 to 29).

These additional DFT calculations support the fact that dubious interpretations may have been given to the conductance features observed on previous nanographene and NPG studies [see Figure R2 and Refs. 33,35]. In these two papers, dI/dV maps obtained in the unoccupied region (positive voltages) also showed clear conductance features confined at the nanopores (voids and double voids). They were attributed to stem from high-energy SAMO (panel (a)) and IPS (panel (b)) states since a good matching with their wave functions was found. However, taking the electronic orbital decay into account, now we have a deeper understanding on how to interpret these conductance features and can therefore assign the experimental dI/dV maps of Fig. R2 to lowest unoccupied molecular orbitals (LUMOs) and conduction bands (CBs) [see Fig. R5 for the electronic orbital decay in double-void and divacancy nanopores, and Supplementary Figs. 25 to 29].

We have added the following paragraphs to convincingly argue the impact and novelty of our work (page 16-17): *"This filtering procedure also reproduces the electronic orbital decay observed for 7-AGNRs¹¹ and nanographenes/NPGs with divacancy²⁶ and double void³⁵ types of pores, validating the generality of this effect (see Supplementary Figs. 21, 25-29)."*

And the following sentence on page 17: *"These decay features clearly explain the characteristics of the simulated LDOS maps and are key to understand and correctly interpret the origin of the experimental STS fingerprints not only of this manuscript (Figures 2 and 3) but also of other NPG and nanographene structures previously reported^{33,35}"*

Therefore, our work complements the previous works by Söde [Ref.11] and Tersoff [Ref.60], extending the orbital decay effect into notched edges in GNRs and closed nanopores (of any size and shape) in GNRs, nanographenes and NPGs.

Specific remarks

Page 3: I do not understand why the authors claim that the explanation of Söde et al. is 'tentative', it is supported by experiment and simulation in real and reciprocal space.

The reviewer is right. Certainly, the word “tentative” was not an appropriate choice and we have rephrased our previous statement to (page 3): “It was **elucidated** that certain orbitals would concentrate more strongly at the edges of the 7-AGNR due to a lack of cancellation of positive and negative regions of the wave function along the edges¹”

We would like to point out that in Söde’s paper the following was claimed: “*Figure 3(b) also reproduces the concentration of the LDOS at the edges of the 7-AGNR that has been reported in previous experimental works [8,27] and is shown in Fig. 4(a). The effect is explained straightforwardly by a lack of cancellation of positive and negative regions of the wave function at the edge of the 7-AGNR*”. In contrast, as we can see in Figure 4 of our manuscript, the orbital decay and confinement into the g-GNR edges or NPG nanopores happens for all frontier orbitals, with or without a cancellation of positive and negative regions of the wave function at the edge of the g-GNRs or nanopores in NPGs. This is also the case for additional simulations that we have performed on GNRs with divacancy or double void types of pores (see Fig. R5 and Supplementary Figs. 25 to 29).

Page 3: It is not clear what a 'wave function morphology' is and also not what this distortion would be. The wave function $\psi(x,y,z)$ is mapped in STS as $|\psi(x,y,z(x,y))|^2$ to correctly simulate the dI/dV map it needs to be evaluated along the $z(x,y)$ contours, but that is not a distortion.

The reviewer is right and the terms morphology and distortion have been removed from the text. Now we refer to a wave function decay into the vacuum and its confinement into the edges of g-GNRs and nanopores in NPGs.

Regarding the second part of the comment, we would like to point out that we have performed constant-height dI/dV maps in the experiment. Therefore, in the dI/dV maps, the wave function $\psi(x,y,z)$ is mapped as $|\psi(x,y,z_0)|^2$, where z_0 corresponds to the

tip-sample separation. To correctly simulate the constant-height dI/dV map, the wave function needs to be evaluated along the (x,y) plane of a fixed z value.

This is now clarified on page 3 in the main manuscript: *“Therefore, dI/dV maps at constant-height mode, which correlate with the squared modulus of the wave functions at a fixed z that corresponds to the tip-sample distance, ask for a detailed analysis of the wave function decay away from the carbon lattice”*.

It is not clear to me if the dI/dV maps are measured with a CO-functionalized tip, in that case the simulation need to be conducted with a p-wave (or s- and p-wave mixture) tip wave function. The simulations are s-wave type so this could be at odds with a CO-tip. This possible ambiguity should be addressed.

The dI/dV maps were measured with a CO-tip (as mentioned in the methods section, page 21 in the main manuscript). For clarity, this is now mentioned also in the Figure panels and along the main text. As requested by Reviewer 1, we also provide additional STS point spectra and dI/dV maps acquired with a metal tip (see point 1 in Reviewer 1).

For a CO-tip, when the tip is “far away” from the g-GNR or NPG, the s-wave orbital is interacting [Ref.31] and the DFT simulations with a s-wave match the experimental results. To further assure this point, we have performed additional dI/dV maps with a metal tip and we obtain the same results as with a CO-tip (see Figure 3 in the main manuscript and Supplementary Figs. 8 and 16). Therefore, additional DFT simulations with a mixture of s- and p-waves are not necessary (see point 2 in Reviewer 1).

REVIEWERS' COMMENTS

Reviewer #1 (Remarks to the Author):

The authors have answered satisfactorily my questions. A definitive control experiment would have been measuring one of the GNR showing edge states due to topology (close to zero bias) and see how the spectroscopic signal evolves with the different setpoints. However, I understand this experiment is out of the scope of this paper and that the answers, as they are, are satisfactory. I recommend for publication as it is.

Reviewer #2 (Remarks to the Author):

All my concerns have been addressed and I recommend publishing the manuscript.

Reviewer #3 (Remarks to the Author):

This work uses Au(111)-supported semiconducting gulf-type GNRs and NPGs as model systems fostering frontier orbitals that appear confined along the edges and nanopores in STS measurements, providing a deep understanding of the frontier orbitals decay and confinement into the vacuum and the edges of GNRs and nanopores in NPGs. The authors explained all questions I raised. I recommend this manuscript for publication in its present form.

Reviewer #4 (Remarks to the Author):

As I mentioned in my previous review, the work presents experiments of high quality and the interpretation using DFT based STM simulations is sound and gives a proper explanation of the experimental observations. Yet, I also maintain, that the novelty of the work is to apply well known concepts of the distance dependence of the STM/STS contrast in pi-electronic systems. The novelty, is that the authors apply these concepts, appropriately and correctly, to porous and edge modulated graphene nanostructures (GRNS). The application of these concepts, which are at play in all GRNS to a specific subset of structure, where strong effects can be present, warrants in my view not a publication in Nat. Commun.

The authors write that they provide "deep understanding of the electronic orbital decay and confinement into the edges of GNRs and nanopores in NPGs (of general character and broad interest to the scientific community)."

The role of orbital decay in pi-electronic materials is well documented (Michael Rohlfing, PRB B 76, 115421, 2007) it is also well known that at large tip distances STM/STS contrast can occur well outside or in-between structures (e.g. Y. J. Dappe, PRB 91, 045427 (2015)).

Although I have many (may be subjective) doubts on the level of novelty of the findings, from a technical point of view the paper is suitable for publication in Nat. Commun.